# Opening and closing of a cryptic pocket in VP35 toggles it between two different RNA-binding modes

Upasana L Mallimadugula[1], Matthew A Cruz[1], Neha Vithani[1], Maxwell I Zimmerman[1], Gregory R Bowman[2]*

[1]Department of Biochemistry & Molecular Biophysics, Washington University School of Medicine, St Louis, United States; [2]Department of Biochemistry & Biophysics and Bioengineering, Perelman School of Medicine, University of Pennsylvania, Philadelphia, United States

## eLife Assessment

This study provides **important** insights into how cryptic pockets play a role in shaping binding preferences of protein-nucleic acid interactions. By combining biochemical assays and state-of-the-art molecular dynamics simulations, mechanism underlying viral protein 35 (VP35) homologs to bind the backbone of double stranded RNA is presented. The evidence is **compelling** for molecular determinants that suggest two different dsRNA binding modes for VP35 and also underscores the evolutionary importance of these pockets.

*For correspondence:
grbowman@seas.upenn.edu

**Competing interest:** The authors declare that no competing interests exist.

**Abstract** Cryptic pockets are of growing interest as potential drug targets, particularly to control protein-nucleic acid interactions that often occur via flat surfaces. However, it remains unclear whether cryptic pockets contribute to protein function or if they are merely happenstantial features that can easily be evolved away to achieve drug resistance. Here, we explore whether a cryptic pocket in the Interferon Inhibitory Domain (IID) of viral protein 35 (VP35) of Zaire ebolavirus aids its ability to bind double-stranded RNA (dsRNA). We use simulations and experiments to study the relationship between cryptic pocket opening and dsRNA binding of the IIDs of two other filoviruses, Reston and Marburg. These homologs have nearly identical structures but block different interferon pathways due to different affinities for blunt ends and backbone of the dsRNA. Simulations and thiol-labeling experiments demonstrate that the homologs have varying probabilities of pocket opening. Subsequent dsRNA-binding assays suggest that closed conformations preferentially bind dsRNA blunt ends while open conformations prefer binding the backbone. Point mutations that modulate pocket opening proteins further confirm this preference. These results demonstrate that the open cryptic pocket has a function, suggesting cryptic pockets are under selective pressure and may be difficult to evolve away to achieve drug resistance.

## Introduction

Cryptic pockets have garnered significant attention, particularly for their potential as drug targets, but it remains unclear whether they play a role in protein function. Cryptic pockets are pockets that are not observed in the experimentally obtained structure of a protein but form due to thermal fluctuations in the native structure (*Bowman and Geissler, 2012*; *Knoverek et al., 2019*). They are an interesting class of protein dynamics as they are increasingly being explored as drug targets (*Horn and Shoichet, 2004*; *Wenthur et al., 2014*). Allosteric modulators that target cryptic pockets provide

many advantages over orthosteric drugs (*Guarnera and Berezovsky, 2020*). Cryptic pockets can provide a means to target proteins that appear to be undruggable due to a lack of potential binding pockets in experimentally-derived snapshots of the protein, as is often the case with protein-nucleic acid interactions (*Meller et al., 2024*). Furthermore, they allow non-competitive regulation, higher specificity due to greater variation of pocket dynamics within protein families than variation in active or functional sites (*Meller et al., 2023*; *Chio et al., 2015*), and the possibility of enhancing and not just inhibiting function (*Hart et al., 2017*). However, successfully drugging a cryptic pocket would create a selective pressure for the organism to evolve protein variants that lack the pocket to achieve drug resistance. If cryptic pockets are happenstantial features that have no functional significance, then evolving them away could be trivial. On the other hand, it could be impossible to evolve away a cryptic pocket if the pocket is an inevitable consequence of the protein's topology. Alternatively, it could be difficult to evolve away from a cryptic pocket if the open state plays a functional role. To explore these possibilities, it would be useful to study the conservation of cryptic pockets across protein variants.

While there are several examples of functional protein dynamics (*Hong et al., 2018*; *Chen et al., 2019*; *Fraser et al., 2009*; *Saavedra et al., 2018*; *Lim et al., 2018*; *Fisher et al., 2022*), studying cryptic pockets and assessing their functional relevance has been challenging. Cryptic pockets have largely been identified serendipitously when inhibitor-bound structures of proteins are solved and show the inhibitor binds in a cryptic pocket (*Allingham et al., 2005*; *Cimermancic et al., 2016*; *Vajda et al., 2018*). While finding such structures proves the existence of cryptic pockets, it does not provide a facile means to quantify the probability of pocket opening or study the effects of sequence variation on pocket opening in the absence of a ligand. Enhanced sampling simulations (*Zimmerman and Bowman, 2015*; *Oleinikovas et al., 2016*; *Limongelli et al., 2010*) and experimental techniques such as room temperature crystallography (*Fraser et al., 2011*), T-jump spectroscopy (*Davis et al., 2017*), T-jump crystallography (*Wolff et al., 2023*), and thiol labeling (*Bowman et al., 2015*) have been used to study protein dynamics, including cryptic pocket opening, and provide an opportunity to assess the conservation of these structural features.

Here, we explore the functional significance of a cryptic pocket that we recently discovered in VP35 protein from Zaire ebolavirus, which is the virus that causes the disease commonly known as Ebola. VP35 plays an essential role in filovirus immune evasion by binding dsRNA that is formed during replication of the viral genome to prevent these nucleic acids from being discovered by host innate immune receptors such as RIG-I and MDA5 (*Basler et al., 2003*; *Hartman et al., 2004*; *Cárdenas et al., 2006*; *Leung et al., 2012*; *Dilley et al., 2017*). The dsRNA binding affinity of the IID is a significant determinant of virulence, making it an appealing therapeutic target (*Woolsey et al., 2019*; *Leung et al., 2010a*). Though a few studies have attempted to find small molecule drugs to target this protein (*Daino et al., 2018*; *Glanzer et al., 2016*; *Brown et al., 2014*), none have resulted in a successful drug discovery campaign. The IID interacts with RNA via flat surfaces that are often difficult to drug, with some going as far as calling these surfaces 'undruggable' (*Hopkins and Groom, 2002*). We discovered a cryptic pocket in VP35 that allosterically controls dsRNA binding, thereby providing a potential means to target this protein (*Cruz et al., 2022*). However, targeting this pocket would be of little utility if the pocket is just happenstance and the virus can easily evolve resistance by acquiring mutations that prevent the pocket from opening without any fitness cost. Therefore, it is important to understand if the cryptic pocket has a functional role that may be under selective pressure and make the evolution of drug resistance more difficult.

To explore the functional relevance of this cryptic pocket, we used a combination of simulations and experiments to study the relationship between cryptic pocket opening and dsRNA binding in the IIDs of two other filoviruses, Reston ebolavirus (Reston) and Marburg marburgvirus (Marburg). Zaire binds to dsRNA blunt ends, thereby blocking RIG-I binding to the RNA. It can also bind to the backbone to block MDA5 binding (*Leung et al., 2010b*; *Edwards et al., 2016*; *Ramanan et al., 2012*). In contrast, Marburg binds the backbone but not the blunt ends, thereby blocking MDA5 binding but is unable to block RIG-I binding to shorter blunt-ended RNAs (*Ramanan et al., 2012*). Reston is known to bind both blunt ends and the backbone of the RNA but is a slightly weaker inhibitor of the interferon response than Zaire (*Leung et al., 2010b*). However, it is difficult to explain these differences given that the crystal structures of all three variants are essentially identical in both their RNA-bound and free forms (*Figure 1*) and all three variants have high sequence similarity in residues directly interacting with dsRNA (*Figure 1—figure supplement 1*; *Ramanan et al., 2012*).

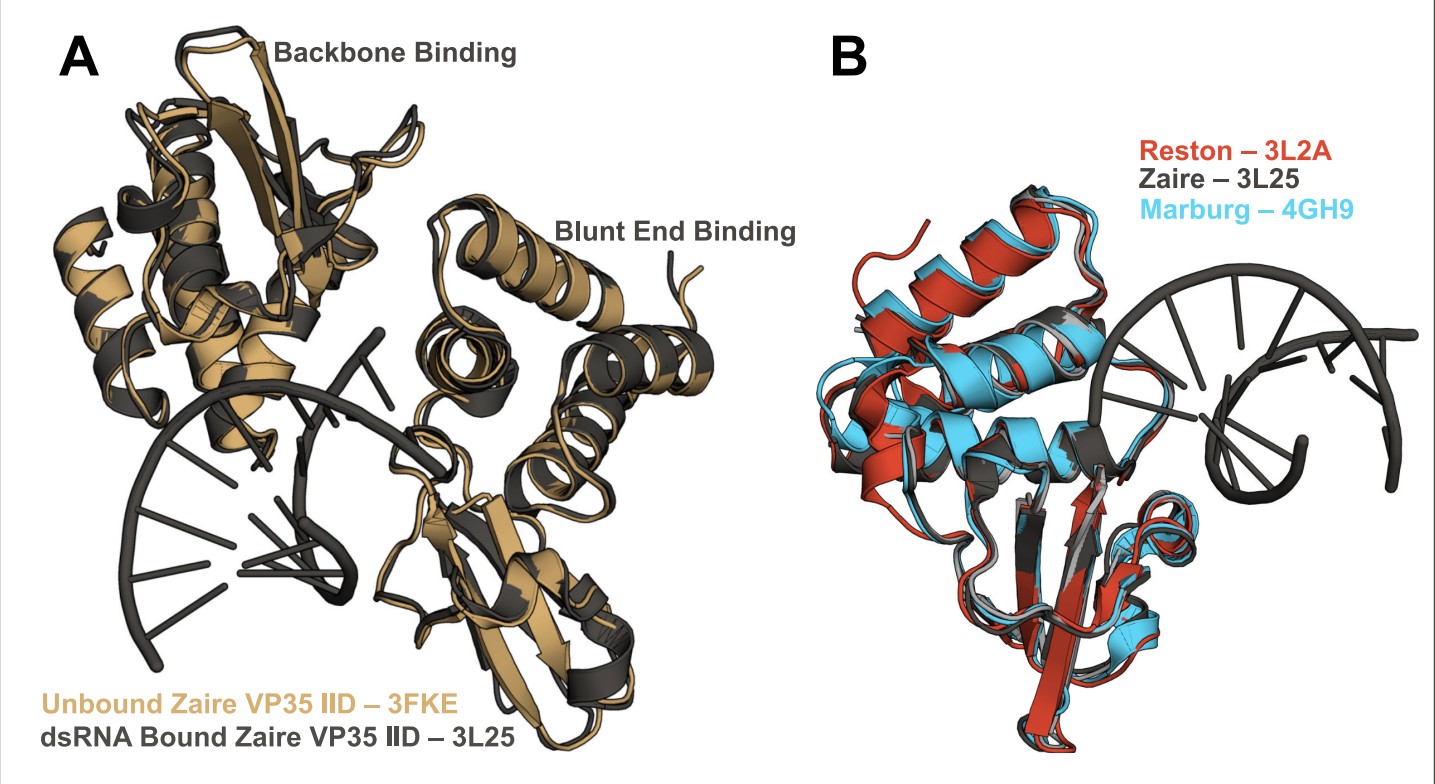

**Figure 1.** Crystal structures of the RNA-bound and unbound states of all three VP35 interferon inhibitory domain s (IIDs) are nearly identical. (**A**) Crystal structures of Zaire IID alone (3FKE, brown) and bound to 8 bp dsRNA (3L25, gray) show that both the blunt end and backbone binding poses of Zaire are nearly identical to the unbound (apo) structure. (**B**) Overlay of Zaire IID bound to 8 bp dsRNA (3L25, gray) with unbound structures of Reston IID (3L2A, red) and Marburg IID (4GH9, blue).

The online version of this article includes the following figure supplement(s) for figure 1:

**Figure supplement 1.** Pairwise sequence comparisons of all homologs used in this study.

Therefore, we hypothesized that differences in cryptic pocket dynamics are responsible for the functional differences between VP35 homologs.

## Results

### Simulations predict the probability of cryptic pocket opening is low in Reston and high in Marburg compared to Zaire

To assess the probability of cryptic pocket opening in the three filoviruses, we first used molecular dynamics simulations guided by adaptive sampling. We first ran 8 µs of simulation of each variant using a goal-oriented adaptive sampling algorithm called FAST (*Zimmerman and Bowman, 2015*) that balances between broad exploration of conformational space and focusing data acquisition on conformations with more open pockets. We then built a Markov State Model (MSM) of each homolog from these simulations and used the centers as seeds for long MD simulations on the Folding@home distributed computing platform (*Voelz et al., 2023*). We collected a total of 126 and 124 µs of data for Reston and Marburg IIDs. Then we built a new MSM for each variant that incorporates the data from Folding@home. The cryptic pocket observed in Zaire IID occurs as the helix spanning residues 305–309 moves away from the alpha helical domain. Therefore, we characterize pocket opening based on the distance between residues 236 and 306 in Zaire, which corresponds to residues 225 and 295 in Reston and Marburg. For the sake of brevity, unless mentioned otherwise, we will use Reston and Marburg numbering going forward. We thus obtain the probability distribution of the distance between residues 225 and 295 based on the equilibrium probability of each structure in the MSM (*Figure 2*).

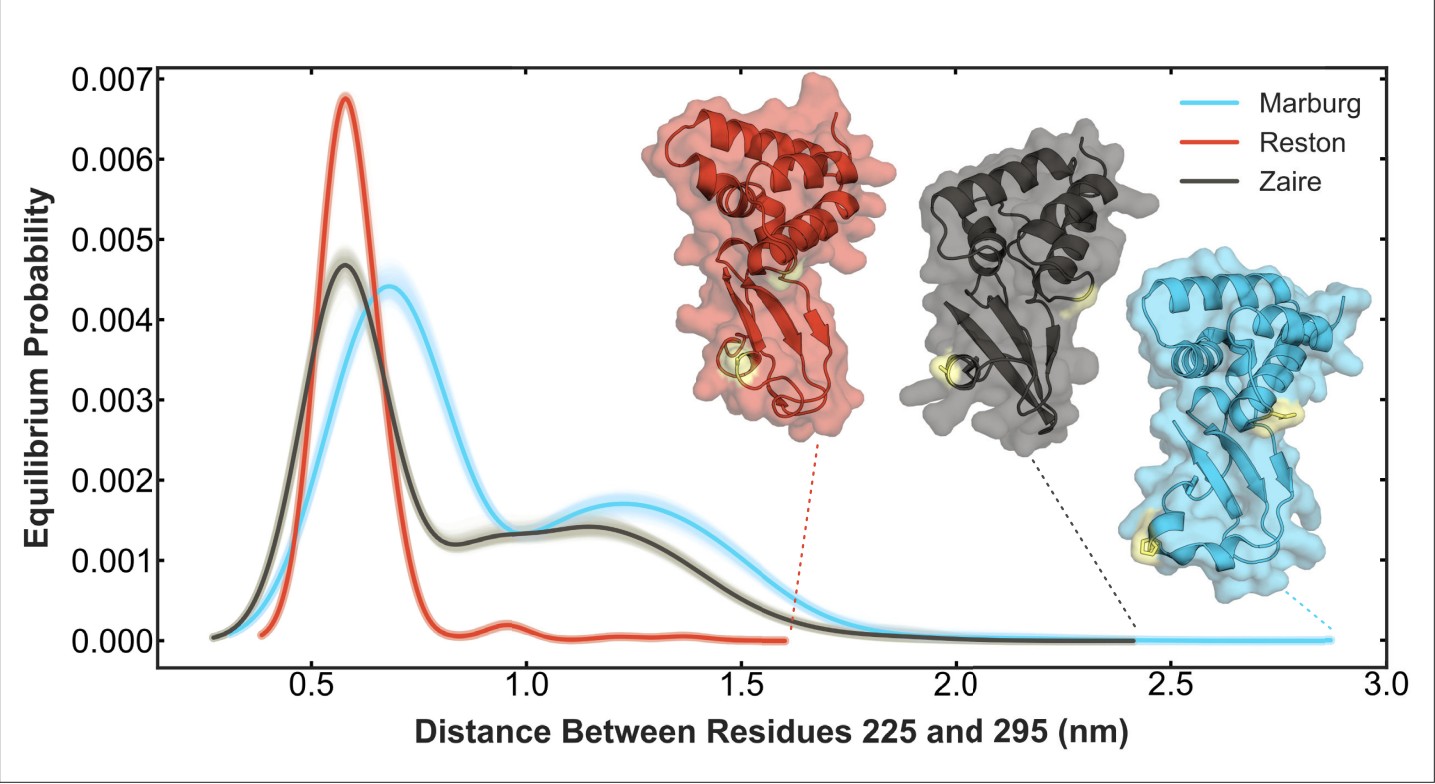

**Figure 2.** Simulations predict the probability of pocket opening is lowest in Reston and highest in Marburg. Each curve is the probability distribution of the distance between two residues that serves as a proxy for pocket opening. Reston VP35 interferon inhibitory domain (IID) (red) shows the least probability of opening the pocket. Zaire VP35 IID shows a greater probability of opening the pocket as well as an increased maximum distance of pocket opening. Marburg VP35 IID shows the highest probability of pocket opening and the highest maximum distance of pocket opening. The shadow around each curve shows the distributions obtained for 100 Markov State Models (MSMs) constructed using random samples of the data chosen with replacement to indicate the statistical variability in the MSM construction. The solid line indicates the mean equilibrium probability of all the bootstraps. The structure with the largest pocket (i.e. largest distance between the two residues) for each variant is shown in ribbon with a transparent surface, using the same color scheme as for the probability distributions. Residues 225 (236 in Zaire) and 295 (306 in Zaire) are shown in yellow sticks.

The online version of this article includes the following figure supplement(s) for figure 2:

**Figure supplement 1.** Structure of Reston and Marburg VP35 interferon inhibitory domains (IIDs) with residues in the allosteric network colored according to the CARDs community they belong to.

**Figure supplement 2.** Implied timescales tests for the six slowest eigenvectors of Markov State Models (MSMs) of Reston interferon inhibitory domain (IID) (**A**) and Marburg IID (**B**).

We find that Marburg has a higher probability of opening than Zaire, but that Reston has a significantly lower probability (*Figure 2*). Furthermore, Marburg opens more widely than the other variants, as judged by reaching larger distances between residues 225 and 295 (*Figure 2* insets). We also performed CARDS analysis (*Singh and Bowman, 2017*) on the Marburg and Reston IID simulations to compare to the allosteric network obtained for Zaire IID using this analysis (*Cruz et al., 2022*; *Figure 2—figure supplement 2*). Similar to Zaire IID, we observe a community of residues around the pocket. We also observe that this community is strongly connected to the green and orange communities that contain residues that interact with dsRNA, indicating that the pocket opening could be affecting dsRNA binding in these homologs as well. However, we also note that the strength of communication both within and between these communities differs between the homologs suggesting that pocket opening has varying influence on the dihedrals of the RNA binding residues in the homologs. Furthermore, the differences in pocket opening have an interesting correspondence to the fact that Marburg prefers to bind the dsRNA backbone, while Reston and Zaire bind both blunt ends and the backbone. We, therefore, hypothesized that open pocket conformations preferentially bind the backbone while closed conformations preferentially bind blunt ends.

## Thiol labeling experiments confirm Marburg has the highest probability of opening and Reston the lowest

As a first test of our computational predictions, we used thiol labeling to experimentally measure the probability that the cryptic pocket is open in each IID homolog. In these experiments, we measure the rate of covalent modification of a cysteine in the cryptic pocket, as we have done previously with β-lactamases (*Knoverek et al., 2021*; *Porter et al., 2019a*) and Zaire IID (*Cruz et al., 2022*). In these experiments, 5,5'-dithiobis-(2-nitrobenzoic acid), or DTNB, is added to the protein sample. In the presence of an oxidized (solvent-exposed) thiol group of a cysteine residue, the disulfide bond between the two TNB molecules breaks, and one TNB molecule attaches to the cysteine via a disulfide bond. The free TNB molecule left from this reaction absorbs light at 412 nm. Therefore, when a cysteine buried inside a pocket is exposed to solvent as the pocket opens, we observe an exponential increase in absorbance at a rate that depends on the opening and closing rates of the pocket. We quantify this using the Linderstrøm-Lang model (see Materials and methods). We focus on thiol labeling of cysteine 296 as its solvent exposure is correlated with the extent of pocket opening (as measured by the distance between residue 225 and 295) in our simulations (*Figure 3A, B and C*), whereas the solvent exposures of other cysteines are not correlated with pocket opening (*Figure 3*, *Figure 3—figure supplement 1*).

As expected from the MSMs, we observe the highest probability of pocket opening in Marburg, followed by Zaire and Reston in that order. Thiol labeling of Reston IID and Marburg IID fit to one exponential per cysteine, each with the same amplitudes, indicating that all the cysteines in each variant get labeled (*Figure 3—figure supplement 2*). We performed point mutations of individual cysteines to serines to assign observed labeling rates to the cysteines as we have previously done for Zaire IID (*Figure 3—figure supplement 3*). We compare these results to the labeling rates we previously observed for Zaire (*Cruz et al., 2022*). The cysteines in Reston's cryptic pocket label far more slowly than those in Zaire, while the labeling rates of the cysteines in Marburg are intermediate between Zaire and Reston (*Figure 3D and H*). These data were fit to the Linderstrøm-Lang model (see Materials and methods) to quantify the probability and kinetics of pocket opening (*Supplementary file 1*). In all three homologs, C296 labels faster than expected from labeling of the unfolded fraction (*Figure 3—figure supplement 4*), suggesting that the pocket does exist in all three homologs. The equilibrium constants for C296 exposure (*Equation 3* in Materials and Methods) for Reston, Zaire, and Marburg IIDs are 0.078±0.001, 0.413±0.009, 5.2±0.5. These correspond to probabilities of pocket opening of 0.072, 0.292, and 0.839 for Reston, Zaire, and Marburg, respectively. Therefore, in agreement with our computational prediction, we observe the highest probability of pocket opening in Marburg, followed by Zaire and Reston in that order.

## Binding to different length RNAs suggests closed conformations preferentially bind dsRNA blunt ends while open conformations prefer binding the backbone

To test our prediction that more open homologs prefer the backbone while more closed ones preferentially bind the blunt ends, we used a fluorescence polarization assay to quantify the affinity of the homologs to different length dsRNA substrates, thereby varying the number of backbone binding sites available. Briefly, we use dsRNA labeled with fluorescein on one 5' end and titrate in varying concentrations of the proteins into a fixed concentration of the RNA (100 nM). Free RNA emits depolarized light upon excitation with polarized light due to its fast rotation, whereas bound RNA emits polarized light. Recording the fluorescence polarization throughout the titration gives us the fraction of RNA bound to the protein. Particularly, we used a short 8 bp RNA to get an accurate measure of binding to the blunt ends and the same 8 bp RNA with a 2-nucleotide overhang on the 3' ends to get an accurate measure of the binding to the backbone alone. Structural studies on these homologs suggest that the backbone binding mode has a footprint of three nucleotides (*Leung et al., 2010a*; *Leung et al., 2010b*; *Ramanan et al., 2012*). Previous RNA binding studies also suggest multiple IID molecules bind to each RNA molecule in solution (*Leung et al., 2010a*; *Edwards et al., 2016*). Therefore, we also performed measurements with a longer 25 bp RNA with and without a 2 nucleotide overhang on the 3' ends to get an accurate measure of any cooperativity in the backbone binding. To accurately account for multiple protein molecules binding to a single molecule of dsRNA, we used a one-dimensional lattice binding model to fit the experimental data (see Materials and methods).

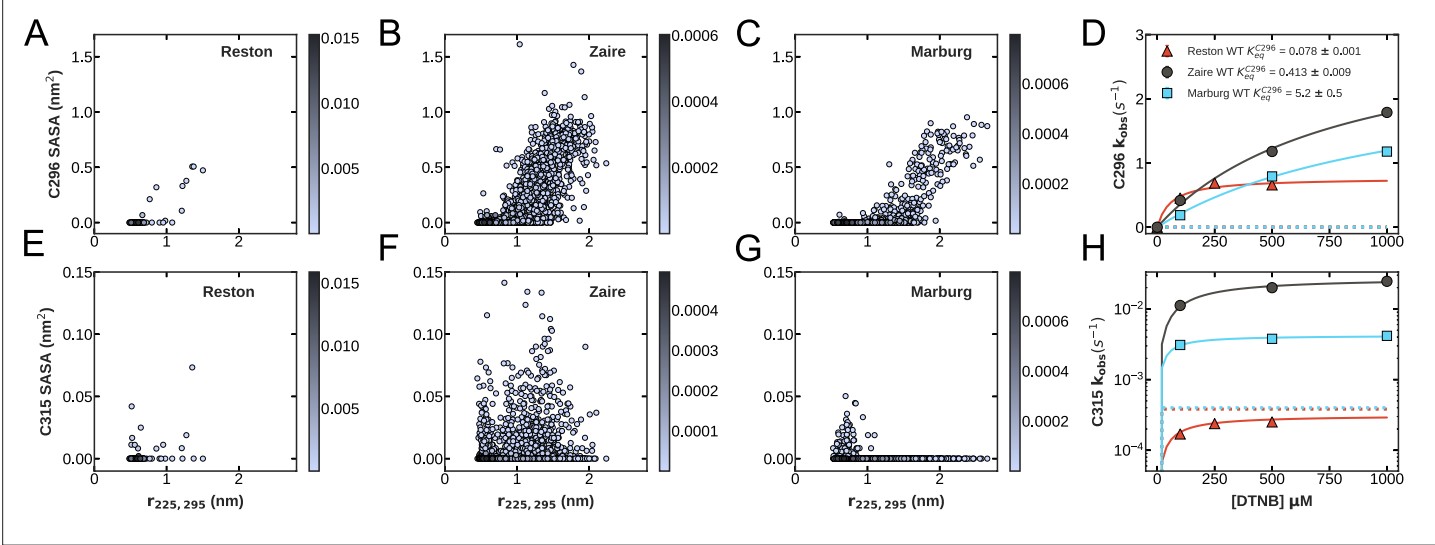

**Figure 3.** Thiol labeling of C296 confirms that the cryptic pocket in Marburg has the highest probability of being open, while Reston has the lowest probability of opening. (**A–C**) Plots of the distance between residues 225 and 295 vs. the Solvent Accessible Surface Area (SASA) of (**A**) C296 of Reston interferon inhibitory domain (IID), (**B**) C307 of Zaire IID, (**C**) C296 of Marburg IID. Each point on the plot represents a Markov State Model (MSM) center and is colored according to its equilibrium probability. (**D**) Observed thiol labeling rates for C296/C307 of Zaire IID (black circles), Marburg IID (blue squares), and Reston IID (red triangles) at a range of 5,5'-dithiobis-(2-nitrobenzoic acid) (DTNB) concentrations. (**E–G**) Plots of the distance between residues 225 and 295 vs. the solvent accessible surface area (SASA) of (**E**) C315 of Reston IID, (**F**) C326 of Zaire IID, and (**G**) C315 of Marburg IID calculated from our MSMs. The SASA of C296 is more correlated with the opening of the cryptic pocket than the SASA of C315, so we focus on thiol labeling of C296 to experimentally characterize pocket opening. (**H**) Observed thiol labeling rates for C315/C326 of Zaire IID (black circles), Marburg IID (blue squares), and Reston IID (red triangles) at a range of DTNB concentrations. Fits to the Linderstrøm–Lang model are shown in colored lines and the expected labeling rate from the unfolded state is shown as blue dotted lines for Marburg IID and red dotted lines for Reston IID. This rate is estimated from the stability and unfolding rate measured for these homologs shown in *Figure 3—figure supplement 4*. The mean and standard deviation from three measurements are shown for D and H. Error bars are smaller than the marker used.

The online version of this article includes the following figure supplement(s) for figure 3:

**Figure supplement 1.** Plots of solvent accessible surface area (SASA) of C264 of Reston IID (**A**) and ZAIRE interferon inhibitory domain (IID) (**B**) and C236 of Reston IID (**C**) and ZAIRE IID (**D**) against the distance between residues 225 and 295 calculated from our Markov State Models (MSMs).

**Figure supplement 2.** Representative time traces from (**A**) three repeats of a thiol labeling experiment (red) performed on Reston interferon inhibitory domain (IID) at 100 μM 5,5'-dithiobis-(2-nitrobenzoic acid) (DTNB) and a quadruple exponential fit (black) and (**B**) one repeat of a thiol labeling experiment (black) performed on MARV IID at 100 μM DTNB and a double exponential fit (red).

**Figure supplement 3.** Kobs vs [5,5'-dithiobis-(2-nitrobenzoic acid), DTNB] plots from thiol labeling of Reston interferon inhibitory domain (IID) wild-type (WT) (**A**), Reston IID C236S/C264S (**B**), Marburg IID WT (**C**), and Marburg IID C296S.

**Figure supplement 4.** Stability of the homologs obtained from a two-state fit to urea denaturation observed via intrinsic tryptophan fluorescence (**A**) Reston wild-type (WT) (light red triangles, solid lines), Reston P280A (dark red triangles, dashed lines), (**B**) Marburg WT.

This model allows nearest-neighbor cooperativity between all binding modes. Globally fitting these four binding curves for each homolog to this model (*Figure 4A, B and C*) generates estimates for the equilibrium constant for opening the pocket (*Figure 4—figure supplement 1A*), the dissociation constants of the closed and open conformations to each site on the backbone ($K_{D,backbone}^{closed}$ and $K_{D,backbone}^{open}$, respectively), the dissociation constant of closed conformations to the blunt end of the RNA ($K_{D,end}^{closed}$), and the cooperativity between these interactions (*Figure 4D*, *Figure 4—figure supplement 1B*).

In support of our proposed model, we find that the equilibrium constant for pocket opening ($K_{oc}$, where oc stands for open vs closed) from our fits to dsRNA-binding data are in good agreement with those from our thiol labeling experiments $\left(K_{eq}^{C296}\right)$. If the pocket opening observed in the thiol-labeling assays is the same conformational change affecting the RNA binding, we would expect the $K_{oc}$ obtained from the fits to be comparable to $K_{eq}^{C296}$. Indeed, $K_{oc}$ for all three homologs obtained

from the fits to the binding model agree very well with the equilibrium constant for the exposure of C296 (C307 in Zaire IID) obtained from the thiol labeling assay (*Figure 4—figure supplement 1A*).

In support of our hypothesis, our global fits show that the probability of pocket opening is the main difference between the variants. As described above, the equilibrium constants for pocket opening vary over a range of about 80-fold between the three homologs. In contrast, each of the dissociation constants between either open or closed protein and the blunt ends or backbone of dsRNA varies by no more than a factor of about sevenfold between the homologs. We also do not observe large differences in the magnitude of the cooperativity parameters between the homologs (*Figure 4—figure supplement 1B*). The interaction between the closed state and the blunt ends of dsRNA is far stronger than the interaction of either the closed state or the open state for the backbone (*Figure 4D*). There is also negative cooperativity between two closed structures binding the backbone at adjacent positions (*Figure 4—figure supplement 1B*). As a result, homologs with low probabilities of pocket opening predominantly bind the blunt ends and their binding curves are left-shifted compared to more open homologs. Homologs with a higher probability of pocket opening are less likely to bind the blunt ends, despite the high affinity of the closed state for blunt ends, since the open state is incompatible with blunt end binding. The interaction between the open state and the backbone ($K_{D, backbone}^{open}$) is about two orders of magnitude stronger than that of the closed state ($K_{D,backbone}^{closed}$) for all three homologs (*Figure 4D*). There is also strong positive cooperativity between a protein in the open state binding the backbone alongside either a closed or open state (*Figure 4—figure supplement 1B*). As a result, homologs with high probabilities of pocket opening, like Marburg, predominantly bind the backbone and have right-shifted binding curves compared to more closed variants. Interestingly, most of the proteins that do bind the backbone are in the open state even for homologs where pocket opening is rare (*Figure 4E and F*). For example, about 80% of the backbone is covered by Reston proteins in the open state even though the equilibrium constant for pocket opening ($K_{oc}$) is 0.061±0.008, which corresponds to a probability of cryptic pocket opening of 0.057, the lowest of the three homologs.

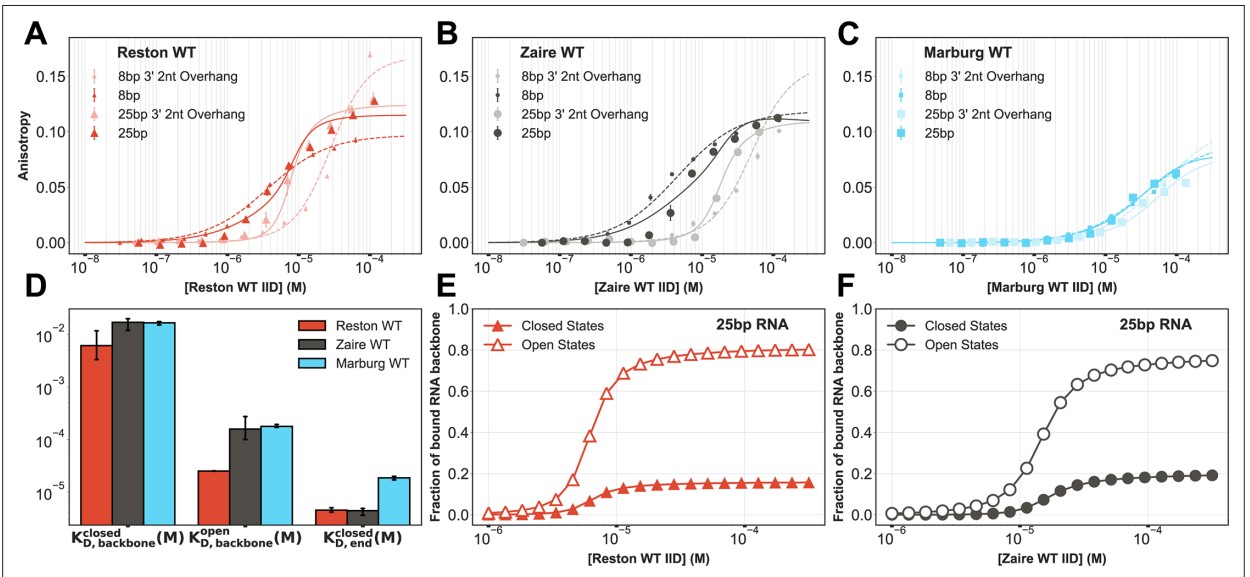

**Figure 4.** Binding to different length dsRNAs suggests closed conformations preferentially bind dsRNA blunt ends while open conformations prefer binding the backbone. (**A–C**) Binding to fluorescently labeled 8 bp (dashed lines) and 25 bp RNA (solid lines) with and without a 3' 2 nucleotide overhang of (**A**) wild-type (WT) Reston interferon inhibitory domain (IID) (**B**) WT Zaire IID (**C**) WT Marburg IID. The anisotropy was measured via a fluorescence polarization assay, converted to anisotropy, and fit to a one-dimensional lattice binding model. The mean and standard deviation from three replicates is shown but error bars are generally smaller than the symbols. (**D**) Comparison of binding affinities obtained from the global fits. The mean and standard deviation from fits to each of the three replicates are shown. (**E–F**) Fraction of the backbone of 25 bp RNA covered by the open states (empty markers) and closed states (colored markers) calculated from the binding parameters obtained from the fits for (**E**) Reston WT IID and (**F**) Zaire WT IID.

The online version of this article includes the following figure supplement(s) for figure 4:

**Figure supplement 1.** Equilibrium constants for pocket opening and cooperativities between binding modes obtained from the global fit to the binding model.

It is also noteworthy that crystal structures capture the closed state bound to the backbone but not the open state, suggesting that the crystallization conditions may shift the equilibrium in favor of the closed state.

## Point mutations that alter the probability of pocket opening also induce differential binding to blunt ends versus the backbone

As a further test of our model, we next sought to identify point mutations that modulate the probability of pocket opening and assess if they have the expected impact on backbone versus blunt end binding. While the sequence of Marburg IID differs significantly from Reston and Zaire IIDs with a sequence identity of 42% and 45%, respectively (*Figure 1—figure supplement 1*), the sequences of Reston and Zaire IID are 88% identical and 94% similar. Particularly, substitutions between these homologs are all distal to the RNA-binding interfaces and all the residues known to make contacts with dsRNA from structural studies are identical. Therefore, we reasoned that comparing these two homologs would help us identify minimal substitutions that control pocket opening probability and allow us to study its effect on dsRNA binding with minimal perturbation of other factors. Of these substitutions, one causes an interesting structural difference between Zaire and Reston IIDs (*Leung et al., 2010b*). Residue P280 in Reston IID results in a formation of an alpha helix, where an alanine in the same structural position (A291) in Zaire IID results in a disordered loop. This disordered loop forms a hinge that rotates as the pocket opens, and we reasoned this motion may be inhibited by the structure the proline induces in Reston. Therefore, we hypothesized that mutating the alanine in Zaire IID to a proline would result in a reduced probability of opening the pocket and a stronger preference for binding dsRNA blunt ends. This hypothesis is supported by our past work showing that the substitution A291P in Zaire IID reduces the probability of pocket opening (*Cruz et al., 2022*). Furthermore, we propose that mutating the proline in Reston to an alanine should increase the probability of pocket opening and enhance the binding to the dsRNA backbone.

As expected, introducing A291P into the Zaire IID leads to a reduced probability of pocket opening in both simulations and thiol labeling experiments, while introducing P280A into the Reston IID leads to an increased probability of pocket opening. FAST simulations performed on these variants indeed showed an increased probability of opening the pocket in Reston IID P280A and a decreased probability in Zaire IID A291P (*Figure 5A*). The pocket also opens more widely in Reston IID P280A than Wild Type (WT) Reston IID. As before, examining the solvent exposure of the cysteines observed in simulation shows that while exposure of C296 (C307 in Zaire) is correlated with pocket opening, that of other cysteines is not (*Figure 5—figure supplement 1*). We followed this up with the thiol labeling assay and assigned labeling rates to cysteines by labeling point mutations of individual cysteines to serines (*Figure 5—figure supplement 2*). We observe that both cysteines in Reston IID P280A labeled faster than WT Reston IID (*Figure 5B*, *Figure 5—figure supplement 3A*). In comparison, both cysteines in Zaire IID A291P labeled slower compared to WT Zaire IID (*Figure 5B*, *Figure 5—figure supplement 3B*). Fits to the Linderstrøm-Lang model show a 26-fold increase in the equilibrium constant for C296 exposure in Reston IID P280A compared to Reston IID WT and a 36-fold decrease in Zaire IID A291P compared to Zaire IID WT (*Figure 5B*). We observed only moderate differences in the stability of Reston IID WT and Reston IID P280A under urea denaturation (*Figure 3—figure supplement 4*). Taken together, this indicates that the mutation modulates pocket opening without significantly impacting the stability of the protein.

In further support of our model, the point mutation that increases Reston's probability of pocket opening also shifts the protein towards backbone binding while the mutation that reduces the probability of pocket opening in Zaire shifts the balance towards blunt end binding. We performed fluorescence polarization binding experiments to the four RNA substrates we used for the WT homologs (*Figure 6A–D*). We analyzed these experiments using global fits of the binding of each mutant to all four RNAs to the one-dimensional lattice model described for the WT homologs (*Figure 6A-D*, *Figure 6—figure supplement 1*). Importantly, the equilibrium constants for pocket opening ($K_{oc}$) obtained from the fits to the binding model continue to agree very well with the equilibrium constants for the exposure of C296 obtained from the thiol labeling assay (*Figure 6E*). While there are large differences between the equilibrium constant for pocket opening amongst these variants, the rest of the fit parameters are very similar to those from the WT proteins. We observe that the dissociation constant of the closed conformations to each site on the backbone

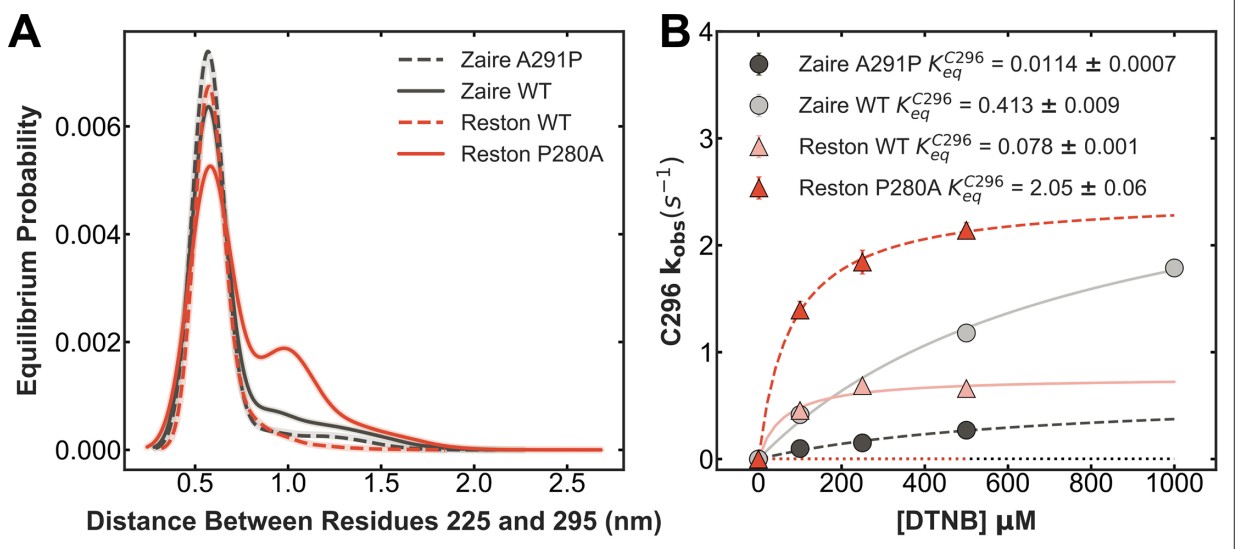

**Figure 5.** Single amino acid substitutions at residue 280 (291 in Zaire) modulate the probability of pocket opening. (**A**) Probability distribution of the distance between residues 225 and 295 obtained from Markov State Models (MSMs) built from FAST adaptive sampling simulations of Reston interferon inhibitory domain (IID) wild-type (WT) (solid red), Reston IID P280A (dashed red), Zaire IID WT (solid black), and Zaire IID A291P (dashed black). The shadow around each curve shows the distributions obtained for 10 MSMs constructed using random samples of the data chosen with replacement to indicate the statistical variability in the MSM construction. The solid line indicates the mean equilibrium probability of all the bootstraps. (**B**) Observed labeling rates of C296 for Reston IID WT (transparent solid red) and Reston IID P280A (dark dashed red) Zaire IID WT (transparent solid black), and Zaire IID A291P (dashed black).

The online version of this article includes the following figure supplement(s) for figure 5:

**Figure supplement 1.** Probability distributions of solvent exposure of Cysteine 296 and Cysteine 315 in Reston P280A and Zaire A291P IIDs.

**Figure supplement 2.** Kobs vs [5,5'-dithiobis-(2-nitrobenzoic acid), DTNB] plots from thiol labeling of Reston interferon inhibitory domain (IID) P280A (**A**), Reston IID P280A/C236S/C264S.

**Figure supplement 3.** Kobs vs [5,5'-dithiobis-(2-nitrobenzoic acid), DTNB] plots for C315 in Reston interferon inhibitory domain (IID) P280A (**A**) and Zaire IID A291P (**B**).

($K_{D,backbone}^{closed}$) is two orders of magnitude weaker than that of the open conformations ($K_{D,backbone}^{open}$) for the mutants as well (**Figure 6F**), further increasing our confidence in the model. We do not observe large differences in the magnitude of the binding and cooperativity parameters between the WTs and the mutants, except for the cooperativity between two neighboring molecules in the open state binding to the backbone $\sigma_{oo}$ Figure (6 F, G, and H). This parameter seems to buffer the effect of the increased affinity of the open state to the backbone. This inherent trade-off also suggests that shifting the equilibrium too drastically in favor of the open conformation would not be beneficial for binding to the backbone. Particularly, Reston P280A is more open than WT Reston, but its binding curves for the overhang substrates are right-shifted compared to WT. This suggests that though the open pocket conformations bind better to the backbone, increasing the fraction of open conformation in solution has a negative effect on the overall backbone binding. To understand this, we calculated the fraction of the RNA backbone covered by the open and closed states of the various homologs and mutants for different lengths of RNA based on the binding parameters obtained from the fit (**Figure 6—figure supplement 2**). We see that in all cases, a much larger proportion of the backbone is covered by the open states than by the closed states. This suggests that while it is necessary to have some ability to open the pocket, increasing the probability of opening has no added benefit for backbone binding. Meanwhile, Zaire A291P is more closed and prefers to bind the blunt ends more than the WT protein. This is reflected by its binding curve for the blunt ended substrates being left-shifted compared to WT. Overall, these results show a strong correspondence between the probability of pocket opening and the relative affinities of the VP35 IID for different binding sites on dsRNA.

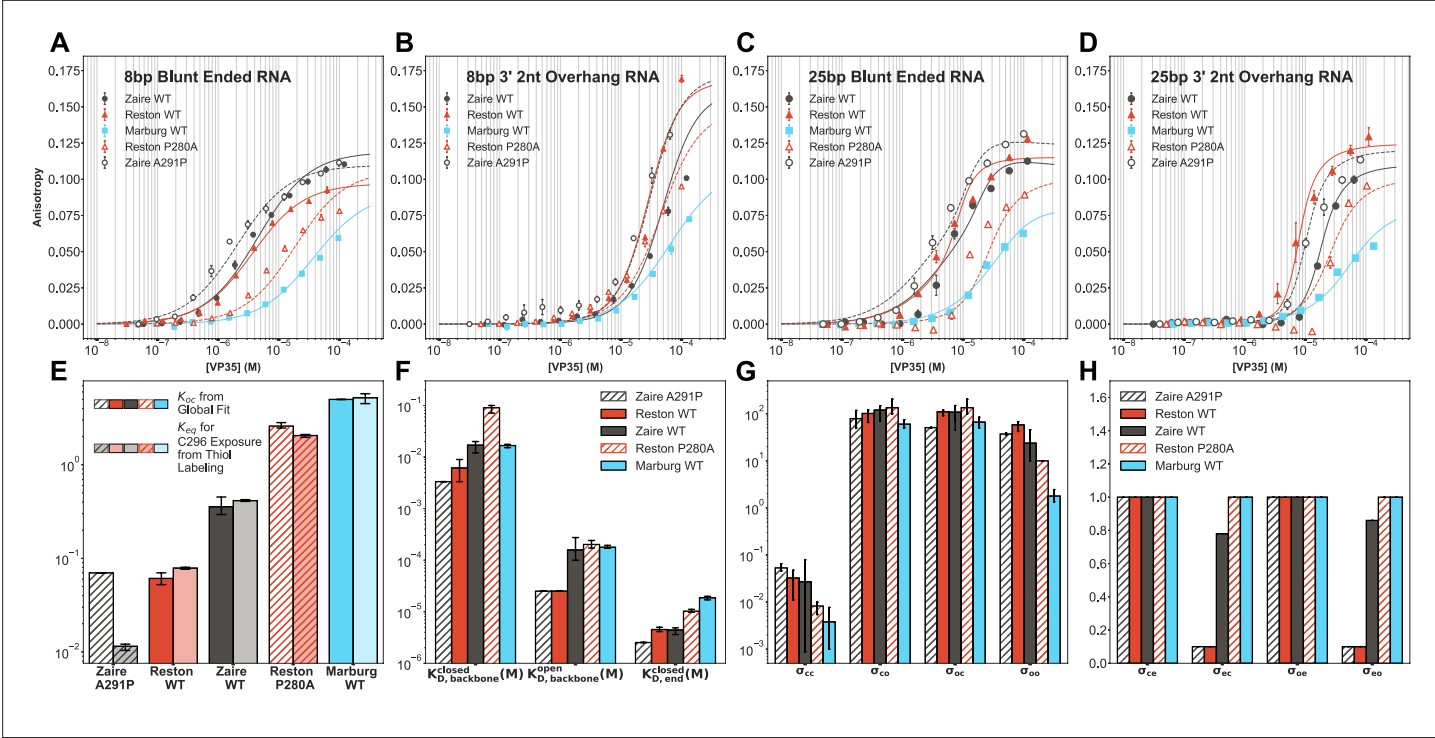

**Figure 6.** Single amino acid substitutions that alter the probability of pocket opening also induce differential binding to blunt ends and the backbone. (A–D) Binding of Zaire A291P interferon inhibitory domain (IID) (empty black circles), wild-type (WT) IID (solid red triangles), WT Zaire IID (solid black circles), Reston P280A IID (empty red triangles), WT Marburg IID (blue squares) to fluorescently labeled (A) 8 bp blunt ended RNA (B) 8 bp RNA with two nucleotide overhangs on 3' ends (C) 25 bp blunt ended RNA (D) 25 bp RNA with two nucleotide overhangs on 3' ends. The anisotropy was measured via a fluorescence polarization assay, converted to anisotropy, fit to a one-dimensional lattice binding model. The mean and standard deviation from three replicates are shown but error bars are generally smaller than the symbols. Lines indicate the global fits. (E) Comparison of $K_{oc}$ obtained from the global fits to $K_{eq}$ for C296 exposure obtained from 5,5'-dithiobis-(2-nitrobenzoic acid) (DTNB) labeling experiments shown in *Figure 3D*. (F) Comparison of dissociation constants obtained from the global fits. (G) Comparison of cooperativity between backbone binding modes obtained from the global fits. (H) Comparison of cooperativity between end and backbone binding modes obtained from the global fits. The mean values of fits to three individual replicates are shown for E, F, G, and H. Standard deviations are shown as error bars.

The online version of this article includes the following figure supplement(s) for figure 6:

**Figure supplement 1.** Global fits to the binding model of (A) Reston P280A interferon inhibitory domain (IID) and (B) Zaire A291P IID.

**Figure supplement 2.** Fraction of the RNA backbone covered by the open and closed states of the various homologs and mutants (in increasing probability of pocket opening from left to right) for 8 bp, 25 bp, and 100 bp blunt ended RNA calculated from the binding parameters obtained from the global fits.

**Figure supplement 3.** Sum of squared residuals for fits of the RNA binding of all five variants of interferon inhibitory domain (IID) used in this study for various combinations of binding site sizes for the three binding modes.

**Figure supplement 4.** Fits and resulting parameters with the backbone binding site size of the closed state of three nucleotides, backbone binding site size of the open state of four nucleotides and the end binding site size as one nucleotide.

**Figure supplement 5.** Fits and resulting parameters with the backbone binding site size of the closed state of four nucleotides, backbone binding site size of the open state of three nucleotides and the end binding site size as one nucleotide.

**Figure supplement 6.** Fits and resulting parameters with the backbone binding site size of the closed state of three nucleotides, backbone binding site size of the open state of three nucleotides and the end binding site size as one nucleotide.

**Figure supplement 7.** Fits and resulting parameters with the backbone binding site size of the closed state of four nucleotides, backbone binding site size of the open state of four nucleotides and the end binding site size as four nucleotides.

**Figure supplement 8.** List of different statistical weights for each base pair in the presence of the open and the closed states.

**Figure supplement 9.** Maximum anisotropy parameters for the various RNAs obtained the global fits of all five variants of the interferon inhibitory domain (IID) used in this study.

## Discussion

We have shown that the opening and closing of a cryptic pocket in VP35 toggles the protein between two different dsRNA binding modes. Specifically, we have shown that VP35 preferentially binds the blunt ends of dsRNA when the cryptic pocket is closed but preferentially binds the backbone when the pocket is open. As a result, variants where the pocket tends to be closed are better at blocking RIG-I from binding dsRNA, whereas VP35 variants that are open more often are better at blocking MDA5. This model explains differences in the RNA binding behaviors of the Zaire, Reston, and Marburg variants of VP35 that were difficult to explain based on crystal structures of the proteins. Moreover, it predicts how point mutations that alter the probability of cryptic pocket opening will alter the relative affinity of a VP35 protein for the backbone versus the blunt ends of dsRNA. More importantly, cryptic pockets are not just happenstance, as they can have a functional role. This finding suggests that cryptic pockets are under selective pressure. As a result, it shouldn't be trivial for viruses or cells to evolve resistance to drugs that target cryptic pockets by acquiring mutations that prevent pocket opening. Future studies on other proteins will help to establish if these findings are generally true or are limited to a subset of cryptic pockets. In the meantime, efforts to target the cryptic pocket in VP35 are warranted given the evidence that this pocket has a functional role that should prevent Zaire ebolavirus from acquiring mutations that give rise to drug resistance by preventing the pocket from opening.

## Materials and methods

### Molecular dynamics simulations and analysis

Simulations for Reston IID and Marburg IID were initiated from the apoprotein models of PDB 3L2A (Reston IID *Leung et al., 2010b*) and 4GHL (Marburg IID *Ramanan et al., 2012*) and run with Gromacs (*Van Der Spoel et al., 2005*) using the amber03 force field (*Duan et al., 2003*) and TIP3P explicit solvent (*Jorgensen et al., 1983*) at a temperature of 300 K and 1 bar pressure, as described previously (*Hart et al., 2016*). We first applied our FAST-pockets algorithm (*Zimmerman and Bowman, 2015*) to replicate the simulation protocol of Zaire IID in *Cruz et al., 2022*. As done previously, we performed ten rounds of FAST simulations with 10 simulations/round and 80 ns/simulation. We then performed an RMSD-based clustering using a hybrid k-centers/k-medoids algorithm (*Beauchamp et al., 2011*) implemented in Enspara (*Porter et al., 2019b*) to divide the data into 1000 clusters. Then we ran three simulations initiated from each cluster center on the Folding@home distributed computing environment, resulting in an aggregate simulation time of 126 μs for Reston IID and 124 μs for Marburg IID. We calculated the Solvent Accessible Surface Area (SASA) of all atoms in each frame using a 2.8 Å probe. We summed up the SASA of all the atoms of the side chain of each residue and clustered this sidechain SASA using k-centers upto a cluster radius of 3.5 nm$^2$ for Reston IID and 2.7 nm$^2$ for Marburg IID followed by five rounds of k-medoids. We then built Markov State Models (MSMs) at multiple lagtimes and used the implied timescales test (*Figure 2—figure supplement 2*) to decide on a 6 ns lag time for the final MSMs of all homologs.

### Protein expression and purification

All variants of VP35's IID were purified from the cytoplasm of *E. coli* BL21(DE3) Gold cells (Agilent Technologies) following the protocols detailed in *Leung et al., 2010b*; *Ramanan et al., 2012*; *Leung et al., 2009*. Variants were generated using the site-directed mutagenesis and confirmed by DNA sequencing. Cells were transformed using heat shock at 42 °C. Transformed cells were grown in LB media (Fisher Scientific or Lambda Biotech, Ballwin, MO) at 37 °C until OD 0.3 then grown at 18 °C until induction at OD 0.6 with 1 mM IPTG (Gold Biotechnology, Olivette, MO). Cells were grown for 15 hr then centrifuged. The pellet was resuspended in 20 mM sodium phosphate pH 8, 1 M sodium chloride, 5 mM Imidazole with 5.1 mM $\beta$-mercaptoethanol. Resuspended cells were subjected to sonication at 4 °C followed by centrifugation. The supernatant was then subjected to Ni-NTA affinity (BioRad Bio-Scale Mini Nuvia IMAC column) and eluted with 20 mM sodium phosphate pH 8, 1 M sodium chloride, 250 mM Imidazole with 5.1 mM $\beta$-mercaptoethanol. This was dialyzed to a final buffer of 20 mM sodium phosphate pH 8, 50 mM NaCl with 5.1 mM $\beta$-mercaptoethanol. The dialyzed sample was subjected to TEV digestion (proTEV plus, Promega) at room temperature for 24–48 hr. We followed this by cation exchange (BioRad UNOsphere Rapid S column) and eluted using a slow

gradient of 20 mM sodium phosphate pH 8, 1 M NaCl with 5.1 mM $\beta$-mercaptoethanol. Cleaved VP35 with the sequence shown in *Figure 1—figure supplement 1* elutes at around 180 mM NaCl. We follow this by size-exclusion chromatography (BioRad Enrich SEC 70 column or Cytiva HiLoad 16/600 Superdex 75) into 10 mM HEPES pH 7, 150 mM NaCl, 1 mM MgCl$_2$, 2 mM TCEP.

## Thiol labeling

We monitored the change in absorbance over time of 5,5'-dithiobis-(2-nitrobenzoic acid) (DTNB, Ellman's reagent, Thermo Fisher Scientific). Various concentrations (100–1000 µM) of DTNB were added to the 5 µM protein and change in absorbance was measured in an SX-20 Stopped Flow instrument (Applied Photophysics, Leatherhead, UK) at 412 nm until the reaction reached a steady state (300–1200 s). Each time course was fit with as many cysteines being labeled for that protein to obtain an observed labeling rate ($k_{obs}$) for each cysteine at each DTNB concentration. The $k_{obs}$ as a function of DTNB concentration for each cysteine were fit with a Linderstrøm–Lang model, shown below, to extract the thermodynamics and/or kinetics of pocket opening, as described in detail previously (*Porter et al., 2019a*).

$$\text{Closed} \underset{k_{close}}{\overset{k_{open}}{\rightleftharpoons}} \text{Open} \xrightarrow{k_{int}[DTNB]} \text{Labeled} \tag{1}$$

Which leads to,

$$k_{obs} = \frac{k_{open} * k_{int}[DTNB]}{k_{close} + k_{open} + k_{int}[DTNB]} \tag{2}$$

From this fit, we report the equilibrium constant for cysteine exposure calculated as:

$$K_{eq} = \frac{k_{open}}{k_{closed}} \tag{3}$$

As a control, the equilibrium constant for folding and the unfolding rate were measured (*Figure 3—figure supplement 4*) and used to predict the expected labeling rate from the unfolded state. The equilibrium constant was inferred from a two-state fit to urea melts monitored by fluorescence and unfolding rates were inferred from exponential fits to unfolding curves monitored by fluorescence after the addition of urea, as described previously (*Bowman et al., 2015*; *Porter et al., 2019a*; *Zimmerman et al., 2017*). Fluorescence data were collected using a Jasco FP-8300 Spectrofluorometer with Jasco ETC-815 Peltier and Koolance Exos2 Liquid Coolant-controlled cuvette holder.

## Fluorescence polarization assay

Binding affinities between homologs and variants of VP35's IID and dsRNA were measured using fluorescence polarization in 10 mM HEPES pH 7, 150 mM NaCl, 1 mM MgCl$_2$. 8 bp, and 25 bp FITC-dsRNA (Integrated DNA Technologies) substrates with or without a 2-nucleotide 3' overhang of the following sequences were included at 100 nM.

| RNA | Sequence |
| --- | --- |
| 8 bp Blunt ended | 5'-/56-FAM/CGCAUGCG-3' <br> 5'-CGCAUGCG-3' |
| 8 bp 2nt 3' Overhang | 5'-/56-FAM/CGCAUGCGCU-3' <br> 5'-CGCAUGCGCU-3' |
| 25 bp Blunt ended | 5'-/56-FAM/AAACUGAAAGGGAGAAGUGAAAGUG-3' <br> 5'-CACUUUCACUUCUCCCUUUCAGUUU-3' |
| 25 bp 2nt 3' Overhang | 5'-/56-FAM/AAACUGAAAGGGAGAAGUGA AAGUGCU-3' <br> 5'-CACUUUCACUUCUCCCUUUCAGUUUCU-3' |

The sample was equilibrated for 1 hr before data collection. Data were collected on a BioTek Synergy2 Multi-Mode Reader as polarization and were converted to anisotropy (r) as in *Keck, 2012*:

$$r = \frac{2p}{3 - p}$$

Where $p$ in the polarization calculated from the observed parallel and perpendicular intensities, $I_{\parallel}$ and $I_{\perp}$ as:

$$p = \frac{I_{\parallel} - GI_{\perp}}{I_{\parallel} + GI_{\perp}}$$

The total fluorescence intensities were checked at each well for fluorescence anomalies such as quenching.

## Analysis of dsRNA binding experiments

To obtain the contributions from the end-binding and backbone-binding to the increase in anisotropy observed in the fluorescence polarization experiments, we used a one-dimensional lattice model for proteins competing to bind to nucleic acids. Here, we consider our labeled dsRNA to be the macromolecule and the IIDs to be the ligands. Specifically, our model has the open and the closed conformations of the IIDs competing for binding to the dsRNA. The total concentration of the ligand is known for every point in the titration and relates to the relative concentration of the open and the closed states as:

$$[IID]_{total} = \left(1 + K_{oc}\right) \left[IID\right]_{closed} \tag{4}$$

$$[IID]_{open} = K_{oc} \left[IID\right]_{closed} \tag{5}$$

To treat the simplest case first, we assume that only the closed state binds to the ends of the dsRNA. We refer to the intrinsic association constant for this interaction as $K_A^{end}$. We assume that the open and closed states bind to the dsRNA backbone with intrinsic association constants $K_A^{open}$ and $K_A^{closed}$.

We performed the fits at varying binding site sizes for the three interactions described above. Specifically, we varied the end binding site size between 1–8 nucleotides and the backbone binding site sizes between 3–8 nucleotides, choosing the lower limits in both cases to be the binding site size observed in the available crystal structures. We analyzed the sum of the residuals squared obtained from these fits across all five variants used in this study to obtain the binding site sizes that produce the best fit (*Figure 6—figure supplement 3*). This analysis showed that the best fit was obtained for the blunt end and both backbone binding site sizes all being three nucleotides each. Using a threshold of five times the minimum sum of the residuals squared, we note that binding site sizes ranging from one to four nucleotides for the end binding, 3–5 nucleotides for the closed state binding to the backbone, and three to four nucleotides for the open state binding to the end produce comparable fits.

Furthermore, we found that the major conclusions of the study hold true for varying binding site sizes in this range (*Figure 6—figure supplement 4*; *Figure 6—figure supplement 5* and *Figure 6—figure supplement 7*). Future RNA binding studies with a technique with a higher sensitivity for binding site sizes will help in confidently estimating these parameters.

The dissociation constants referred to in the text are inverses of the association constants described above. Therefore,

$$K_{D,\,backbone}^{closed} = \frac{1}{K_A^{closed}} \tag{6}$$

$$K_{D,\,backbone}^{open} = \frac{1}{K_A^{open}} \tag{7}$$

$$K_{D,\,end}^{closed} = \frac{1}{K_A^{end}} \tag{8}$$

We treat the backbone binding of the closed state, the backbone binding of the open state, and the end-binding of the closed state as three ligand species competing for binding the dsRNA. We denote the ligand type by $s = 1, 2 \ or \ 3$ for each of those cases, respectively.

We use the transfer matrix method detailed in **Nilsson et al., 2014** to calculate the probability, $p_s(i)$, that a base pair $i$ is occupied by a ligand of type $s$ given by:

$$p_s(i) = \frac{Z_s(i)}{Z} \tag{9}$$

Where $Z$ is the partition function and $Z_s(i)$ is a sum over all allowed Boltzmann-weighted states consistent with base pair $i$, for $i \in (1, N)$ where N is the length of the dsRNA, being covered by a type $s$ ligand. $Z_s(i)$ and $Z$ are calculated using transfer matrices.

To construct the transfer matrix for each nucleotide $i$, we enumerate all the possible states a given nucleotide $i$ can be in and assign each state a statistical weight according to the method in **Nilsson et al., 2014**. We need to make two major changes to this model to describe our system.

The first is to account for the fact that the bulk concentrations of the ligand species in our system are not independent. In the model, the statistical weight for the binding of ligand of type $s$ is given by $c_s K_s$ where $c_s$ is the bulk concentration of the ligand and $K_s$ is its intrinsic association constant. In our system, the bulk concentrations of the ligands are the concentration of the open and closed conformations of the protein. These concentrations are related as shown in **Equations 4; 5**. Therefore, the statistical weights for each of the interactions in our system are given by $K_{eff}^s$ where,

$$K_{eff}^1 = \frac{K_A^{closed}[IID]_{total}}{(1 + Koc)} \tag{10}$$

$$K_{eff}^2 = \frac{Koc\, K_A^{open}[IID]_{total}}{(1 + Koc)} \tag{11}$$

$$K_{eff}^3 = \frac{\left(K_A^{end} - K_A^{closed}\right)[IID]_{total}}{(1 + Koc)} \tag{12}$$

The second change we need to make is to account for the fact that end-binding can only occur at nucleotides $i = 1\ or\ N - N_{end} + 1$, where $N_{end}$ is the end binding site size. Therefore, the value of $K_{eff}^3$ given by **Equation 12** is only included in transfer matrices for nucleotides $1\ or\ N - N_{end} + 1$ and we set $K_{eff}^3 = 0$ for all other nucleotides.

Furthermore, we allow cooperativity between all binding modes denoted by the constants $\sigma_{ss'}$ where $s$ and $s'$ can be $1, 2\ or\ 3$ denoting the open state binding to the backbone, closed state binding to the backbone and the closed state binding to the ends, respectively. For example, the cooperativity between one closed conformation binding to the backbone and an open conformation binding to the backbone to its right is given by $\sigma_{12}$. For ease of understanding, we refer to ligand types s = 1, 2 and 3 as o, c, and e in the text. Therefore, the cooperativity between one closed conformation binding to the backbone and an open conformation binding to the backbone to its right is given by $\sigma_{co}$. Examples of these statistical weights are shown in **Figure 6—figure supplement 8**.

With these statistical weights, we construct the transfer matrices for each nucleotide $i$ as below, with $K_{eff}^3 = 0$ for all nucleotides other than 1 and $N - N_{end} + 1$.

$$\begin{pmatrix}
1 & 0 & 0 & 0 & 0 & 0 & 1 & 0 & 0 & 1 \\
1 & 0 & 0 & \sigma_{11} & 0 & 0 & \sigma_{12} & 0 & 0 & \sigma_{13} \\
0 & 1 & 0 & 0 & 0 & 0 & 0 & 0 & 0 & 0 \\
0 & 0 & K_{eff}^1 & 0 & 0 & 0 & 0 & 0 & 0 & 0 \\
1 & 0 & 0 & \sigma_{21} & 0 & 0 & \sigma_{22} & 0 & 0 & \sigma_{23} \\
0 & 0 & 0 & 0 & 1 & 0 & 0 & 0 & 0 & 0 \\
0 & 0 & 0 & 0 & 0 & K_{eff}^2 & 0 & 0 & 0 & 0 \\
1 & 0 & 0 & \sigma_{31} & 0 & 0 & \sigma_{32} & 0 & 0 & 0 \\
0 & 0 & 0 & 0 & 0 & 0 & 0 & 1 & 0 & 0 \\
0 & 0 & 0 & 0 & 0 & 0 & 0 & 0 & K_{eff}^3 & 0
\end{pmatrix}$$

We use these to calculate the probability, $p_s(i)$, that a base-pair $i$ is occupied by a ligand of type $s$ given by **Equation 9**.

From this, we calculate the average probability of the backbone being bound as

$$p_b = \frac{\sum_{i=1}^{N} \left( p_1(i) + p_2(i) \right)}{N} \tag{13}$$

and the average probability of the ends being bound as

$$p_{end} = \frac{p_3(1) + p_3(N)}{2} \tag{14}$$

We then convert this average probability to observed anisotropy ($r_{obs}$) as follows:

$$r_{obs} = \left( r_0 - r_{max}^{end} \right) p_{end} + \left( r_0 - r_{max}^{b} \right) p_b \tag{15}$$

Where, $r_o$ is the anisotropy of the free RNA.

$r_{max}^{end}$ is the maximum anisotropy for blunt end binding,

$r_{max}^{b}$ is the maximum anisotropy for backbone binding,

Here, $r_{max}^{end}$ and $r_{max}^{b}$ are parameters obtained from the fit and are shown in **Figure 6—figure supplement 9**.

Fits were performed using SciPy 1.8.0, NumPy 1.22.2, and lmfit 1.2.2.

## Acknowledgements

We thank Dr. Andrea Soranno for numerous helpful discussions. We are grateful to the citizen scientists who participate in Folding@home for volunteering to run simulations on their personal computers. We thank Dr. Gaya Amarasinghe for providing the plasmids for Reston and Marburg IIDs. This work was supported by NIH NIGMS R35GM152085 and NSF MCB 2218156. MAC was supported by the NIH grants 5R25GM103757 to WUSTL IMSD program, and NIH F31AI157079.

## Additional information

### Funding

| Funder | Grant reference number | Author |
| --- | --- | --- |
| National Institute of General Medical Sciences | R35GM152085 | Gregory R Bowman |
| NSF Division of Molecular and Cellular Biosciences | 2218156 | Gregory R Bowman |
| National Institutes of Health | F31AI157079 | Matthew A Cruz |

The funders had no role in study design, data collection and interpretation, or the decision to submit the work for publication.

### Author contributions

Upasana L Mallimadugula, Conceptualization, Data curation, Formal analysis, Investigation, Methodology, Project administration; Matthew A Cruz, Neha Vithani, Maxwell I Zimmerman, Conceptualization, Data curation, Formal analysis, Investigation, Methodology; Gregory R Bowman, Conceptualization, Funding acquisition, Investigation, Methodology, Project administration

### Author ORCIDs

Upasana L Mallimadugula ⓘ https://orcid.org/0000-0002-4269-3541

Gregory R Bowman ⓘ https://orcid.org/0000-0002-2083-4892

Reviewer #1 (Public review): https://doi.org/10.7554/eLife.104514.3.sa1

Reviewer #2 (Public review): https://doi.org/10.7554/eLife.104514.3.sa2

Reviewer #3 (Public review): https://doi.org/10.7554/eLife.104514.3.sa3

Author response https://doi.org/10.7554/eLife.104514.3.sa4

## Additional files

### Supplementary files

Supplementary file 1. Parameters for the Linderstrøm-Lang model obtained from fits of the Thiol-labeling experiments.

MDAR checklist

### Data availability

The molecular dynamics datasets that support this study are available at https://zenodo.org/records/15854842. MSM data, MD starting structures, and experimental source data have been deposited in the Open Science Framework database (https://osf.io/t245v). MSM data used for Zaire IID is available on OSF at https://osf.io/5pg2a. Source code for FAST, CARDS, and Enspara (MSM building and analysis software; *Zimmerman, 2023*, *Porter and Zimmerman, 2025*) are available on GitHub at https://github.com/bowman-lab.

The following datasets were generated:

| Author(s) | Year | Dataset title | Dataset URL | Database and Identifier |
|---|---|---|---|---|
| Mallimadugula UL | 2025 | Opening and closing of a cryptic pocket in VP35 toggles it between two different RNA-binding modes | https://doi.org/10.5281/zenodo.15854842 | Zenodo, 10.5281/zenodo.15854842 |
| Mallimadugula UL | 2025 | Opening and closing of a cryptic pocket in VP35 toggles it between two different RNA-binding modes | https://osf.io/t245v | Open Science Framework, t245v |

The following previously published dataset was used:

| Author(s) | Year | Dataset title | Dataset URL | Database and Identifier |
|---|---|---|---|---|
| Cruz M, Bowman G, Zhang S | 2022 | A cryptic pocket in Ebola VP35 allosterically controls RNA binding | https://osf.io/5pg2a/ | Open Science Framework, 5pg2a |

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
