## [Editor Report · eLife Assessment]

This study provides **important** insights into how cryptic pockets play a role in shaping binding preferences of protein-nucleic acid interactions. By combining biochemical assays and state-of-the-art molecular dynamics simulations, mechanism underlying viral protein 35 (VP35) homologs to bind the backbone of double stranded RNA is presented. The evidence is **compelling** for molecular determinants that suggest two different dsRNA binding modes for VP35 and also underscores the evolutionary importance of these pockets.

---

## [Referee Report · Reviewer #1 (Public review)]

Summary:

Mallimadugula et al. combined Molecular Dynamics (MD) simulations, thiol-labeling experiments, and RNA-binding assays to study and compare the RNA-binding behavior of the Interferon Inhibitory Domain (IID) from Viral Protein 35 (VP35) of Zaire ebolavirus, Reston ebolavirus, and Marburg marburgvirus. Although the structures and sequences of these viruses are similar, the authors suggest that differences in RNA binding stem from variations in their intrinsic dynamics, particularly the opening of a cryptic pocket. More precisely, the dynamics of this pocket may influence whether the IID binds to RNA blunt ends or the RNA backbone.

Overall, the authors present important findings to reveal how the intrinsic dynamics of proteins can influence their binding to molecules and, hence, their functions. They have used extensive biased simulations to characterize the opening of a pocket which was not clearly seen in experimental results - at least when the proteins were in their unbound forms. Biochemical assays further validated theoretical results and linked them to RNA binding modes. Thus, with the combination of biochemical assays and state-of-the-art Molecular Dynamics simulations, these results are clearly compelling.

Strengths:

The use of extensive Adaptive Sampling combined with biochemical assays clearly point to the opening of the Interferon Inhibitory Domain (IID) as a factor for RNA binding. This type of approach is especially useful to assess how protein dynamics can affect its function.

Weaknesses:

Although a connection between the cryptic pocket dynamics and RNA binding mode is proposed, the precise molecular mechanism linking pocket opening to RNA binding still remains unclear.

---

## [Referee Report · Reviewer #2 (Public review)]

Summary:

The authors aimed to determine whether a cryptic pocket in the VP35 protein of Zaire ebolavirus has a functional role in RNA binding and, by extension, in immune evasion. They sought to address whether this pocket could be an effective therapeutic target resistant to evolutionary evasion by studying its role in dsRNA binding among different filovirus VP35 homologs. Through simulations and experiments, they demonstrated that cryptic pocket dynamics modulate the RNA binding modes, directly influencing how VP35 variants block RIG-I and MDA5-mediated immune responses.

The authors successfully achieved their aim, showing that the cryptic pocket is not a random structural feature but rather an allosteric regulator of dsRNA binding. Their results not only explain functional differences in VP35 homologs despite their structural similarity but also suggest that targeting this cryptic pocket may offer a viable strategy for drug development with reduced risk of resistance.

This work represents a significant advance in the field of viral immunoevasion and therapeutic targeting of traditionally "undruggable" protein features. By demonstrating the functional relevance of cryptic pockets, the study challenges long-standing assumptions and provides a compelling basis for exploring new drug discovery strategies targeting these previously overlooked regions.

Strengths:

The combination of molecular simulations and experimental approaches is a major strength, enabling the authors to connect structural dynamics with functional outcomes. The use of homologous VP35 proteins from different filoviruses strengthens the study's generality, and the incorporation of point mutations adds mechanistic depth. Furthermore, the ability to reconcile functional differences that could not be explained by crystal structures alone highlights the utility of dynamic studies in uncovering hidden allosteric features.

Weaknesses:

While the methodology is robust, certain limitations should be acknowledged. For example, the study would benefit from a more detailed quantitative analysis of how specific mutations impact RNA binding and cryptic pocket dynamics, as this could provide greater mechanistic insight. This study would also benefit from providing a clear rationale for the selection of the amber03 force field and considering the inclusion of volume-based approaches for pocket analysis. Such revisions will strengthen the robustness and impact of the study.

Comments on revisions:

The authors addressed the concerns raised.

---

## [Referee Report · Reviewer #3 (Public review)]

Summary:

The authors suggest a mechanism that explains the preference of

viral protein 35 (VP35) homologs to bind the backbone of double stranded RNA versus blunt ends. These preferences have a biological impact in terms of the ability of different viruses to escape the immune response of the host.

The proposed mechanism involves the existence of a cryptic pocket, where VP35 binds the blunt ends of dsRNA when the cryptic pocket is closed and preferentially binds the RNA double stranded backbone when the pocket is open.

The authors performed MD simulation results, thiol labelling experiments, fluorescence polarization assays, as well as point mutations to support their hypothesis.

Strengths:

This is a genuinely interesting scientific questions, which is approached through multiple complementary experiments as well as extensive MD simulations. Moreover, structural biology studies focused on RNA-protein interactions are particularly rare, highlighting the importance of further research in this area.

Weaknesses:

- Sequence similarity between Ebola-Zaire (94% similarity) explains their similar behaviour in simulations and experimental assays. Marburg instead is a more distant homolog (~80% similarity relative to Ebola/Zaire). This difference is sequence and structure can explain the propensities, without the need to involve the existence of a cryptic pocket.

- No real evidence for the presence of a cryptic pocket is presented, but rather a distance probability distribution between two residues obtained from extensive MD simulations. It would be interesting to characterise the modelled RNA-protein interface in more detail

Comments on revisions:

-I still think that the term cryptic pocket is misleading here, unless the cryptic pocket is more thoroughly characterised. I would find it more appropriate to use the term open/closed state.

- Mg ions are known to be crucial in stabilising RNA structure both in vitro and in MD simulations (see e.g. Draper BJ 2008 and many others). While I understand that the authors cannot repeat simulations in presence of ions, I believe that this detail should be more clearly detailed in the manuscript.

---

## [Author Response]

The following is the authors’ response to the original reviews

**Public Reviews:**

**Reviewer #1 (Public review):**
Summary:Mallimadugula et al. combined Molecular Dynamics (MD) simulations, thiol-labeling experiments, and RNA-binding assays to study and compare the RNA-binding behavior of the Interferon Inhibitory Domain (IID) from Viral Protein 35 (VP35) of Zaire ebolavirus, Reston ebolavirus, and Marburg marburgvirus. Although the structures and sequences of these viruses are similar, the authors suggest that differences in RNA binding stem from variations in their intrinsic dynamics, particularly the opening of a cryptic pocket. More precisely, the dynamics of this pocket may influence whether the IID binds to RNA blunt ends or the RNA backbone.Overall, the authors present important findings to reveal how the intrinsic dynamics of proteins can influence their binding to molecules and, hence, their functions. They have used extensive biased simulations to characterize the opening of a pocket which was not clearly seen in experimental results - at least when the proteins were in their unbound forms. Biochemical assays further validated theoretical results and linked them to RNA binding modes. Thus, with the combination of biochemical assays and state-of-the-art Molecular Dynamics simulations, these results are clearly compelling.Strengths:The use of extensive Adaptive Sampling combined with biochemical assays clearly points to the opening of the Interferon Inhibitory Domain (IID) as a factor for RNA binding. This type of approach is especially useful to assess how protein dynamics can affect its function.Weaknesses:Although a connection between the cryptic pocket dynamics and RNA binding mode is proposed, the precise molecular mechanism linking pocket opening to RNA binding still remains unclear.
**Reviewer #2 (Public review):**
Summary:The authors aimed to determine whether a cryptic pocket in the VP35 protein of Zaire ebolavirus has a functional role in RNA binding and, by extension, in immune evasion. They sought to address whether this pocket could be an effective therapeutic target resistant to evolutionary evasion by studying its role in dsRNA binding among different filovirus VP35 homologs. Through simulations and experiments, they demonstrated that cryptic pocket dynamics modulate the RNA binding modes, directly influencing how VP35 variants block RIG-I and MDA5-mediated immune responses.The authors successfully achieved their aim, showing that the cryptic pocket is not a random structural feature but rather an allosteric regulator of dsRNA binding. Their results not only explain functional differences in VP35 homologs despite their structural similarity but also suggest that targeting this cryptic pocket may offer a viable strategy for drug development with reduced risk of resistance.This work represents a significant advance in the field of viral immunoevasion and therapeutic targeting of traditionally "undruggable" protein features. By demonstrating the functional relevance of cryptic pockets, the study challenges long-standing assumptions and provides a compelling basis for exploring new drug discovery strategies targeting these previously overlooked regions.Strengths:The combination of molecular simulations and experimental approaches is a major strength, enabling the authors to connect structural dynamics with functional outcomes. The use of homologous VP35 proteins from different filoviruses strengthens the study's generality, and the incorporation of point mutations adds mechanistic depth. Furthermore, the ability to reconcile functional differences that could not be explained by crystal structures alone highlights the utility of dynamic studies in uncovering hidden allosteric features.Weaknesses:While the methodology is robust, certain limitations should be acknowledged. For example, the study would benefit from a more detailed quantitative analysis of how specific mutations impact RNA binding and cryptic pocket dynamics, as this could provide greater mechanistic insight. This study would also benefit from providing a clear rationale for the selection of the amber03 force field and considering the inclusion of volume-based approaches for pocket analysis. Such revisions will strengthen the robustness and impact of the study.
**Reviewer #3 (Public review):**
Summary:The authors suggest a mechanism that explains the preference of viral protein 35 (VP35) homologs to bind the backbone of double-stranded RNA versus blunt ends. These preferences have a biological impact in terms of the ability of different viruses to escape the immune response of the host.The proposed mechanism involves the existence of a cryptic pocket, where VP35 binds the blunt ends of dsRNA when the cryptic pocket is closed and preferentially binds the RNA double-stranded backbone when the pocket is open.The authors performed MD simulation results, thiol labelling experiments, fluorescence polarization assays, as well as point mutations to support their hypothesis.Strengths:This is a genuinely interesting scientific question, which is approached through multiple complementary experiments as well as extensive MD simulations. Moreover, structural biology studies focused on RNA-protein interactions are particularly rare, highlighting the importance of further research in this area.Weaknesses:- Sequence similarity between Ebola-Zaire (94% similarity) explains their similar behaviour in simulations and experimental assays. Marburg instead is a more distant homolog (~80% similarity relative to Ebola/Zaire). This difference is sequence and structure can explain the propensities, without the need to involve the existence of a cryptic pocket.- No real evidence for the presence of a cryptic pocket is presented, but rather a distance probability distribution between two residues obtained from extensive MD simulations. It would be interesting to characterise the modelled RNA-protein interface in more detail
**Recommendations for the authors:**

**Reviewer #1 (Recommendations for the authors):**
Before assessing the overall quality and significance of this work, this reviewer needs to specify the context of this review. This reviewer's expertise lies in biased and unbiased molecular dynamics simulations and structural biology. Hence, while this reviewer can overall understand the results for thiol-labeling and RNA-binding assays, this review will not assess the quality of these biochemical assays and will mainly focus on the modelling results.Overall, the authors present important findings to reveal how the intrinsic dynamics of proteins can influence their binding to molecules and, hence, their functions. They have used extensive biased simulations to characterize the opening of a pocket which was not clearly seen in experimental results - at least when the proteins were in their unbound forms. Biochemical assays further validated theoretical results and linked them to RNA binding modes. Thus, with the combination of biochemical assays and state-of-the-art Molecular Dynamics simulations, these results are clearly compelling.Beyond the clear qualities of this work, I would like to mention a few points that may help to better contextualize and rationalize the results presented here.- First, both the introduction and discussion sections seem relatively condensed. Extending them to, for example, better describe the methodological context and discuss the methodological limitations and potential future developments related to biased simulations may help the reader get a better idea of the significance of this work.- The authors presented 3 homologs in this study: IIDs of Reston, Zaire, and Marburg viruses. While Zaire and Reston are relatively similar in terms of sequence (Figure S1). The sequences clearly differ between Marburg and the two other viruses. Can the author indicate a similarity/identity score for each sequence alignment and extend Figure S1 to really compare Marburg sequence with Reston and Zaire? Can they also discuss how these differences may impact the comparison of the three IIDs? This may also help the reader to understand why sometimes the authors compare the three viruses and why sometimes they are focusing only on comparing Zaire and Reston.

We would like to thank the reviewer for raising this point and we agree that additional details about the sequence comparison provide more context for the choices of substitutions we made. Therefore, we have updated Fig S1 to include a detailed pairwise comparison of all the IID sequences including the percentage sequence similarity and identity. We have also added the following sentences to the results section where we first introduced the substitutions between Zaire and Reston IIDs

“While the sequence of Marburg IID differs significantly from Reston and Zaire IIDs with a sequence identity of 42% and 45% respectively (Fig S1), the sequences of Reston and Zaire IID are 88% identical and 94% similar. Particularly, substitutions between these homologs are all distal to the RNA-binding interfaces and all the residues known to make contacts with dsRNA from structural studies are identical. Therefore, we reasoned that comparing these two homologs would help us identify minimal substitutions that control pocket opening probability and allow us to study its effect on dsRNA binding with minimal perturbation of other factors.”

- In this work, the authors mentioned the cryptic pocket but only illustrated the opening of this pocket by using a simple distance between residues (Figure 2) and a SASA of one cysteine (Figure 3). In previous work done by the authors (Cruz et al. , Nature Communications, 2022), they better characterized residues involved in RNA binding and forming the cryptic pocket. Thus, would it be possible to better described this cryptic pocket (residues involved, volume, etc ..) and better explain how, structurally speaking, it can affect RNA binding mode (blunt ends vs backbone) ?

We thank the reviewer for pointing out the need for clarification on the residues involved in RNA binding and pocket opening and the mechanism linking them. We have performed the CARDS analysis on Reston and Marburg IID simulations as we had done on Zaire IID simulations in Cruz et al, 2022. The results are shown in Fig S3 and discussed in the main text in the first results section.

- As a counter-example, the authors used C315 for SASA calculation and thiol labeling (Figure 3). This cysteine is mainly buried as seen by SASA for Reston and Marburg and thiol labelling (Figure 3 E,G,H). Would it be possible to also get thiol labeling rates for Cystein 264 in Reston and its equivalent to see a case where the residue is solvent exposed?

We have shown the SASA for C264 from the simulations in Fig S4 and the thiol labeling rates for all 4 cysteines in Reston IID in Fig S6. Comparing these rates to the rates of all 4 cysteines obtained for Zaire IID (Fig 4 in Cruz et Al, 2022), we observe that the rates for C264, which is expected to be exposed are significantly faster than those of C315 which is largely buried in all variants.

- I strongly support here the will of the authors to share their data by depositing them in an OSF repository. These data help this reviewer to assess some of the results produced by the authors and help to better understand the dynamics of their respective systems. I have just a few comments that need to be addressed regarding these data: o While there are data for WT Reston and Marburg, there is no data for Zaire. Is this because these data correspond to the previous work (Cruz et al. 2022) (in this case, it would be good to make this clear in the main text) or is it an omission? o There is no center.xtc file in the Marburg-MSM directory o There is no protmasses.pdb in the Reston-MSM directory- In general, if possible, it would be good to use the same name for each type of file presented in each directory to help a potential user understand a bit more how to use these data.- If possible, adding a bit more of metadata and explanations on the OSF webpage would be very beneficial to help find these data. To help in this direction, the authors may have a look to the guidelines presented at the end of this article: https://elifesciences.org/articles/90061

We thank the reviewer for pointing out the omissions from the OSF repository. We have added the missing files and followed a uniform naming convention. We have also added documentation in the metadata section of the OSF repository to help others use the data.

Indeed, the simulation data used for Zaire IID is available on the OSF repository corresponding to Cruz et al. 2022 at https://osf.io/5pg2a. We have also clarified this in the data availability section of the main text.

Minor point:In Figure 2, there is a slight bump for the 225-295 distance around 1 nm for Reston. Can the author comment it ? As these results are based on long AS, even if very small, do the authors think this population is significant?

Comparing the probability distributions obtained from bootstrapping the frames used to calculate the MSM equilibrium probabilities (Revised Fig1), we observe that the bump for the Reston IID distribution is persistent in all bootstraps indicating that it might indeed be significant. This is also consistent with our observation that the cysteine 296 does get fully labeled in our thiol labeling experiments, albeit significantly slowly compared to the other homologs.

**Reviewer #2 (Recommendations for the authors):**
I recommend that the authors implement moderate revisions prior to the publication of this research article, addressing the identified weaknesses (see below).The authors should provide a rationale for their selection of the amber03 force field (Duan et al., JCTC 24, 1999-2012, 2003) for molecular dynamics simulations, particularly given the availability of more recent and optimized versions of the AMBER force fields. These newer force fields may offer improved parameterization for biomolecular systems, potentially enhancing the accuracy and reliability of the simulation results.

We chose the Amber03 force field because it has performed well in much of our past work, including the original prediction of the cryptic pocket that we study in this manuscript. The results presented in this manuscript also demonstrate the predictive power of Amber03.

Additionally, while the authors utilized solvent-accessible surface area (SASA) for cryptic pocket analysis, volume-based approaches may be more suitable for this purpose. Several studies (e.g., Sztain et al. J. Chem. Inf. Model. 2021, 61, 7, 3495-3501) have demonstrated the utility of volume analysis in identifying and characterizing cryptic pockets. The authors could consider incorporating such methodologies to provide a more comprehensive assessment of pocket dynamics.The authors propose that the cryptic pocket is not merely a random structural feature but functions as an allosteric regulator of dsRNA binding. To further substantiate this claim, an in-depth analysis of this allosteric effect using for instance network analysis could significantly enhance the study. Such an approach could identify key residues and interaction networks within the protein that mediate the allosteric regulation. This type of mechanistic insight would not only provide a stronger theoretical framework but also offer valuable information for the rational design of therapeutic interventions targeting the cryptic pocket.

We thank the reviewer for pointing out the need for clarification on the molecular mechanism linking the opening of the cryptic pocket to RNA binding. We have performed the CARDS analysis on Reston and Marburg IID simulations as was done on Zaire IID simulations in Cruz et al, 2022. The results are shown in Fig S3 and discussed in the main text in the first results section. Briefly, we do find a community (blue) comprising the pocket residues in Reston and Marburg IIDs as we did in Zaire. Similarly, we find that many of the RNA binding residues fall into the orange and green communities as in Zaire. However, there are differences in exactly which residues are clustered into which of these two communities. There are also differences in how strongly connected these communities are in the three homologs. Therefore, while we can conclude that pocket residues likely have varying influence on the RNA binding residues in the homologs, it is hard to say exactly what that variation is from this analysis alone.

**Reviewer #3 (Recommendations for the authors):**
- MD simulations: All simulations were initialised from the 3 crystal structures, is it correct? In all cases, RNA ds was not included in simulations, right? Were crystallographic MG ions in the vicinity of the binding site included? these are known to influence structural dynamics to a large extent.

All simulations were indeed initialized using only protein atoms from the crystal structures 3FKE, 4GHL, and 3L2A. Therefore, crystallographic Mg ions were not included in the simulations. However, we do agree with the reviewer and think that the effect of parameters such as salt concentration, specifically Mg ions which are known to be important for the stability of dsRNA, on the pocket opening equilibrium merits detailed study in future work.

- Figure 2: Would it be possible to perform e.g. a block error analysis and show the statistical errors of the distributions?

We agree that showing the statistical variation in the MSM equilibrium probabilities is important for comparing the different distributions. Therefore, we have updated Figs 2 and 5 to show the distributions obtained from MSMs constructed using 100 and 10 random samples of the data respectively to indicate the extent of the statistical variability in the MSM construction.

- More detailed structural biology experiments (such as NMR or HDX-MS) could potentially shed more light on the differential behaviour of the three different homologs, providing more evidence for the presence of the cryptic pocket.

We agree that NMR and HDX-MS are powerful means to study dynamics and are actively exploring these approaches for our future work.